# Scalable Solution-Processed Fabrication Approach for High-Performance Silver Nanowire/MXene Hybrid Transparent Conductive Films

**DOI:** 10.3390/nano11061360

**Published:** 2021-05-21

**Authors:** Pengchang Wang, Chi Zhang, Majiaqi Wu, Jianhua Zhang, Xiao Ling, Lianqiao Yang

**Affiliations:** Key Laboratory of Advanced Display and System Applications, Ministry of Education, Shanghai University, Yanchang Road 149, Shanghai 200072, China; wangpc@shu.edu.cn (P.W.); zhangchi303145@163.com (C.Z.); wumjq@shu.edu.cn (M.W.); jhzhang@staff.shu.edu.cn (J.Z.); ling_xiao@shu.edu.cn (X.L.)

**Keywords:** silver nanowire, Ti_3_C_2_T_x_ MXene, patch, solution-processed, transparent conductive film

## Abstract

The transparent conductive films (TCFs) based on silver nanowires are expected to be a next-generation electrode for flexible electronics. However, their defects such as easy oxidation and high junction resistance limit its wide application in practical situations. Herein, a method of coating Ti_3_C_2_T_x_ with different sizes was proposed to prepare silver nanowire/MXene composite films. The solution-processed silver nanowire (AgNW) networks were patched and welded by capillary force effect through the double-coatings of small and large MXene nanosheets. The sheet resistance of the optimized AgNW/MXene TCFs was 15.1 Ω/sq, the optical transmittance at 550 nm was 89.3%, and the figure of merit value was 214.4. Moreover, the AgNW/MXene TCF showed higher stability at 1600 mechanical bending, annealing at 100 °C for 50 h, and exposure to ambient air for 40 days. These results indicate that the novel AgNW/MXene TCFs have a great potential for high-performance flexible optoelectronic devices.

## 1. Introduction

Over the years, transparent conductive films (TCFs) play an indispensable role in the development of new-generation electronic devices such as organic light-emitting diodes, light-emitting diodes, solar cells, photodetectors, touch screens, electronic papers, and field-effect transistors [1]. As a critical part of flexible electronic products, the flexible transparent films need to have both high optical transparency to visible light and electrical conductivity under repeated mechanical deformation [2]. Indium tin oxide (ITO) has been widely used for transparent conductive films as the conventional conductive metal oxide thin film. Nevertheless, despite its excellent conductivity and transmittance, a few drawbacks of ITO limit its application for flexible electronic devices, such as high cost due to indium scarcity, complicated processing requirements, sensitivity to acidic and basic environments, and cracking when subjected to bending [3,4]. Therefore, it is necessary to develop new materials to replace ITO. Several candidate materials have been developed to prepare flexible TCFs, including conductive polymers [5], metal nanowires [6], metal grids [7,8], ultrathin metal films [9], carbon nanotubes (CNTs) [10], and graphene [11].

Among these materials, silver nanowire (AgNW) is an ideal material for fabricating flexible TCFs due to the high photoelectric performance and good flexibility [4,12]. However, AgNW still has some shortcomings in practical applications. There are many large insulating voids in the conductive network, which limit the conductivity of AgNW film and fundamentally limit the performance of optoelectronic devices [13,14]. In addition, the further development of AgNW TCFs is hindered by low adhesion, high contact resistance at the wire junctions, and poor stability [15,16]. Previous studies have proposed various methods of manufacturing, coating, and post-treating for silver nanowire TCFs [15,17,18]. Among them, AgNW hybrid TCFs are considered to be an attractive method, and many materials have been developed to prepare hybrid TCFs to improve photoelectric performance, stability, adhesion, and flatness [19,20]. Inspired by the directional collection of water droplets on wetted spider silks, Liao et al. claimed that one-dimensional nanomaterials-based transparent films are expected to be directionally patched at the joints to achieve high electric-optical and electric-mechanical performance [21].

In 2011, a new type of two-dimensional transition-metal carbide and nitride, named MXene, was discovered. Due to its distinctive physical and chemical properties, MXene has been extensively studied in numerous applications such as transparent films, ion batteries, supercapacitors, and electromagnetic shielding [22,23,24]. Ti_3_C_2_T_x_ is the most popular one because of its excellent performance. Through selective etching and post-treatment, few or even single layers with a thickness of several nanometers and lateral size of several hundred nanometers of Ti_3_C_2_T_x_ can be obtained, and T is the surface functional groups such as oxygen (O), hydroxyl (OH), and/or fluorine (F) [25,26]. These active end groups give Ti_3_C_2_T_x_ good hydrophilicity and are responsible for controlling the work function [27]. Furthermore, Ti_3_C_2_T_x_ has good electrical properties and optical transmittance, which presents a sheet resistance of 0.5–8 kΩ/sq when the transmittance is between 40 and 90% [16]. These excellent properties of MXene provide more possibilities for obtaining high-performance AgNW-based TCFs. Tang et al. reported that solution-processed MXene/AgNW-PUA transparent electrodes were used for flexible organic solar cells [28]. This method of directly covering MXene and graphene on the silver nanowire network cannot effectively solve the wire junction problem, it requires additional welding steps which limited the large-scale applications usually [29,30]. Therefore, it is urgently needed to prepare high-performance TCFs by adjusting the structures, composition, and preparation methods.

In this work, highly transparent and conductive silver nanowires/MXene hybrid TCFs were prepared by an all-solution process consisting of three spin coating steps. Firstly, silver nanowires were prepared into a conductive network by spin-coating and few-layer MXene nanosheets of different sizes were spin-coated on the conductive network in turn. The small MXene sheets trapped along the nanowires could directionally patch the nanowires and welded the wire junctions via the capillary force effect. The large MXene sheets filled the voids of the network provides more conductive paths which further improve the electrical properties of TCF. The preparation strategy effectively reduced the sheet resistance and roughness of the film. Moreover, due to the abundant functional groups of MXene, the work function and the adhesion between the film and the substrate were improved. Additionally, the MXene can act as a protective layer to prevent the nanowires from being oxidized. The MXene layers improved the environmental stability and mechanical stability of the hybrid film. Therefore, the hybrid TCF can make up for the defects of the silver nanowire-based film and improve its performance, which shows great application prospects in transparent and flexible devices.

## 2. Experimental Section

### 2.1. Materials

All chemicals and reagents in this work were acquired from commercial sources unless otherwise specified. The average diameter of silver nanowires dispersed in isopropanol was ~50 nm and the length was ~40 μm, which were purchased from Sigma-Aldrich (St. Louis, MO, USA). Several layers of Ti_3_C_2_T_x_ MXene dispersed in deionized water (DI water) with a concentration of 5 mg·mL^−1^ were purchased from Jilin 11 Technology Co., Ltd (Changchun, China).

### 2.2. Preparation of the Silver Nanowire/MXene Hybrid Film

First, glass and polyethylene naphtholate (PEN) substrates (2.5 × 3 cm^2^) were immersed in acetone, ethanol, and deionized water for 10 min of ultrasonic cleaning to make the surface clean enough. Subsequently, the substrates were dried by nitrogen and pretreated with UV–ozone-oxidation for 15 min to generate hydrophilic surfaces. The AgNW solution was diluted to a concentration of 0.5 to 5 mg·mL^−1^ with isopropanol. The 2 mg/mL Ti_3_C_2_T_x_ MXene dispersion was ultrasonicated for 1 h, then centrifuged at 3500 rpm for 45 min and 8000 rpm for 15 min. After that, the upper suspension of small Ti_3_C_2_T_x_ nanosheets with few layers was obtained. In order to obtain large MXene nanosheets, a 5 mg/mL Ti_3_C_2_T_x_ MXene dispersion was ultrasonicated for 30 min and centrifuged at 3500 rpm for 15 min. Then, the obtained upper Ti_3_C_2_T_x_ dispersion was diluted with DI water to a volume radio of 1:1 to 1:20 and ultrasonically treated for 30 min.

The AgNWs solution was spin-casted on the substrates at 2000 rpm for 30 s. After annealing at 120 °C for 2 min to volatilize the solvent, the small MXene nanosheets dispersion was spin-coated on the silver nanowire network at 2000 rpm for 1 min, and dried in the air (AgNW/MXene1). Afterwards, large nanosheets were covered on the surface of the hybrid film by continuous spin coating. Additionally, then the composite film was placed in the air to dry naturally (AgNW/MXene2).

### 2.3. Characterization

The transmittance of the TCFs was measured by the UV-vis spectrophotometer (U-3900H. Japan). The sheet resistance of the TCF was determined by the Hall Measurement System (ACCENT HL5550LN2, Columbus, OH, USA). Optical images of the composite films were presented through the optical microscope (KEYENCE, VHX5000. Osaka, Japan). The field-emission scanning electron microscope (FE-SEM, Zeiss Sigma 500, Jena, Germany) was used to obtain SEM images and the elements were analyzed by energy dispersive spectrometer (EDS, Oxford Ultim max, Oxford, UK). The surface roughness of TCF was performed by Atomic force microscopy (AFM, BRUKER, Billerica, MA, USA). X-ray photoelectron spectroscopy (XPS, Thermo Fisher Scientific K-Alpha, Waltham, MA, USA) was used to test chemical elements. The work function of TCFs was measured using a UPS analyzer (VG Scienta R4000. Hollington, UK). A 3M Scotch tape with 12.7 mm width was used to verify the adhesion between the film and the substrate. The water contact angle was measured by a Contact Angle Instrument (Vino, SL200KS. Boston, MA, USA). The resistance of the samples was measured with a VICTOR VC890 D digital multimeter (Hongkong, China).

## 3. Results and Discussion

In this work, the AgNW/MXene transparent conductive films were prepared by an all-solution process, as schematically illustrated in Figure 1a. The commercially AgNW was spin-casted on the surface of the pretreated substrates to form well-organized electrically conductive networks (Figure 1b). After that, few layers of Ti_3_C_2_T_x_ MXene dispersion were spin-casted continuously onto the networks to fabricate hybrid TCFs. Figure 1c showed the XRD pattern of MXene, the disappearance of the (104) peak at ~39°, and the shifting of the (002) peak to 5.8° fully demonstrates the formation of few layers Ti_3_C_2_T_x_ [31]. The Tyndall effect is light scattering by particles in a colloid or a very fine suspension when the particle size is smaller than the wavelength of the incident light. The inset image showed the Tyndall scattering phenomenon of small MXene dispersions and diluted large MXene dispersions, which confirmed that MXene forms a uniform colloidal dispersion without aggregation [32].

To study the influence of the concentration of silver nanowire on its optoelectrical properties, the sheet resistance and transmittance of the TCFs were measured, as shown in Figure 2a. With the increase in the concentration of silver nanowires, the sheet resistance of the AgNW TCFs decreased from 3598.2 to 5.4 Ω/sq, and the transmittance decreased from 98.5% to 82.2%. As a comparison, the sheet resistance and transmittance of the AgNW/MXene1 TCFs with a corresponding concentration were also displayed in Figure 2a. It indicated that the electrical performance of the inherent AgNW TCFs was improved after compounding a certain amount of MXene. Especially when the concentration of silver nanowires was low, the sheet resistance decreased significantly. However, as the concentration of silver nanowires increased, the positive effect of adding MXene to the AgNW network became insignificant. For the transmittance, it was an interesting result that the optical properties of AgNW/MXene hybrid TCFs with small MXene nanosheets were very close to those of AgNW films with the same concentration. In Figure 2b, the SEM image showed the surface morphology of AgNW/MXene1 films, and it was found that the MXene nanosheets were tightly wrapped around the nanowires, especially at the wire junctions. The Zeta-potentials of the AgNW and the Ti_3_C_2_T_x_ MXene were −7.8 and −29 mV (at pH 7), respectively [27,32]. The MXene nanosheets were preferentially captured by the silver nanowires due to the electrostatic attraction between the nanosheets and nanowires. As a result, MXene nanosheets patches at the wire junctions can significantly lower the contact resistance. Additionally, the inset graph showed the welding effect at the junctions, which could be explained by the capillary force effect during solvent evaporation [33]. As predicted by Kelvin’s equation, the solvent water prefers to condense at these nanogaps, which formed positions with large mean curvature [34,35]. The strong capillary force was generated at junctions, and the wire junctions were welded during subsequent water evaporation [33]. These results proved that compared with the pristine AgNW films, the AgNW/MXene1 hybrid films exhibited lower sheet resistance and close transparency at the same concentration. In our work, silver nanowires with a concentration of 3 mg/mL were selected for the following research.

Another critical defect for the AgNW film is low coverage, which may not only make an unfavorable electrical contact but also limit the ‘the performance of devices. However, the AgNW network was not completely covered by the first-time coated small MXene nanosheets. Therefore, the second spin coating of MXene with a higher concentration filled the void space in the AgNW network. To further improve the photoelectric properties of the TCFs, the electrical and optical performance of the TCFs with a constant AgNW concentration and various MXene concentrations were compared. As shown in Figure 2c, the conductivity of hybrid TCFs with second-coating MXene had been improved, the sheet resistance revealed a downward trend and decreased to 11.5 Ω/sq, and the transmittance decreased gradually (Figure 2d). Nevertheless, with the continuous increase in MXene content, the sheet resistance started to increase. It proved that after the MXene content exceeds a certain value, the higher the content of MXene, the higher the resistance. This is because AgNW is the main contributor to the conductivity of TCFs, whose conductivity is much higher than that of MXene.

To characterize and analyze the composition of the transparent conductive film, XPS was used to characterize the film as shown in Figure 3a. The band at a binding energy region of 366–376 eV was attributed to the Ag 3d XPS bands from the AgNWs. It showed that the film surface mainly consisted of Ag, Ti, C, O, the XPS bands and peaks of these elements were very similar to those reported previously [36,37]. SEM and EDS analysis were further used to study the surface morphology of the film and were presented in Figure 3b and Appendix A. The elemental analysis indicated that the silver, carbon, and titanium components in the TCF had a uniform distribution. These results confirmed that MXene nanosheets were uniformly covered on the surface of the AgNW network. According to the bridge effect, the presence of the MXene nanosheets throughout the AgNW network could provide additional electrical paths along with the electrical flow and thereby increase the conductivity of the film (Figure 3c) [38]. The luminescence experiment of LEDs confirmed the improvement of the electrical performance of the hybrid film (Appendix A).

In general, the sheet resistance and transmittance of AgNW-based TCF mutually restrict each other. To quantify the properties, the figure of merit (FoM), which is defined as the ratio of electronic ($\sigma_{DC}$) conductivity to optical conductivity ($\sigma_{OP}$), was introduced to characterize the relationship between optical transmittance and sheet resistance [2,39,40]. The higher the FoM, the better the TCFs. The equation is:(1)$$FoM=\frac{\sigma_{DC}}{\sigma_{OP}}=\frac{188.5}{2R_{sh}\left(T^{-\frac{1}{2}}-1\right)}$$
where $\sigma_{DC}$*,*
$\sigma_{OP}$*,*
$R_{sh}$, *T* are direct current conductivity, optical conductivity, sheet resistance, and optical transmittance (λ = 550 nm) of the film, respectively. As shown in Appendix A, the highest FoM value of AgNW/MXene2 was 214.4 when the sheet resistance is 15.1 Ω/sq, and the transmittance is 89.3% at 550 nm. Also, the transmittance spectra of AgNW, AgNW/MXene1, and AgNW/MXene2 TCFs and the photograph of the different films on glass were showed in Appendix A. In contrast, it was higher than that of the same concentration of large MXene nanosheet directly coated on the silver nanowire network (FoM = 174.9). Subsequently, the photoelectric properties of the TCFs were compared with other reported TCFs, and is presented in Figure 3d. It indicated that the AgNW/MXene TCF exhibited a relatively lower sheet resistance with higher transmittance [13,41,42,43,44,45,46,47].

For TCFs of optoelectronic devices, surface roughness is also a key parameter. Irregularities and protrusions on the film surface can cause short circuits and eventually lead to high leakage currents. After coating by two-dimensional material MXene, the Ti_3_C_2_T_x_ nanosheets were evenly tiled on the nanowire network and filled the network gaps, which could overcome the problems caused by uneven electric fields and stacking of pristine AgNW film. This uniform filling and covering could reduce the surface roughness. To further quantify the variation of surface roughness, the AFM test was performed, and the results are shown in Figure 3e,f. It is observed that the surface roughness of AgNW/MXene2 was 11.5 nm, which was lower than that of AgNW (28.5 nm). As a result, the hybrid TCF had higher optoelectronic performance and further improved the unfavorable morphology. Moreover, the AgNW/MXene TCFs with this structure exhibited much better performance than that of the pristine films in many aspects.

To investigate the surface wetting state of different films, similar conditions were maintained to measure the water contact angles, and the results are displayed in Figure 4a. The contact angle of the pure MXene film is 19.3 degrees, showing good hydrophilicity. After the introduction of MXene, the water contact angle of the silver nanowire film decreased from 51.5 degrees to 30.5 degrees due the active functional groups such as oxygen (O), hydroxyl (OH), and/or fluorine (F). Moreover, due to the abundant surface functional groups, one main potential feature of 2D MXene material is the possibility to adjust its electronic structure, such as work function or band gap [48,49]. UPS was conducted to calculate the work function of the transparent conductive films (Figure 4b). Compared with the pristine silver nanowire film, the work function of the AgNW/MXene film increased by 0.5 eV.

Considering the practical application of films in electronic devices, the TCF should meet certain mechanical requirements to ensure stability. Adhesion on substrates is important for the liquid cleaning and easy handling of AgNW TCFs. A tape test was carried out to evaluate the adhesion of the AgNW film with or without the MXene. A piece 3M Scotch tape was pressed against the surface of the film without air bubbles and then peeled it off perpendicularly to the surface. As shown in Figure 4c,d, the morphology of the film was observed with an optical microscope before and after the tape adhesion test. The stickiness of the tape makes the AgNW film easy to fall off. However, for composite films, the difference in surface morphology before and after the test is relatively small. For this reason, it displayed that the adhesion between the film and the substrate was enhanced after coating MXene.

To further evaluate and quantify the adhesive strength between the substrate and AgNW film with and without MXene, 180° peel measurements were conducted [50,51,52]. A 3M transparent tape was attached to the film tightly and subsequently peeled off the substrate at 180°; the corresponding experimental details were illustrated in the insert image. During this process, the peeling force applied to separate the tape from the substrate was measured. Figure 4e showed the force–displacement curves of samples with and without MXene, and the tested samples were prepared on a glass and PEN substrate. For the PEN substrate, the average peeling force of the AgNW film was 90.5 N/m, which was 74.8% of AgNW/MXene film. The average peeling force of the AgNW/Mxene film on the glass substrate reached 151.3 N/m, which was 1.34 times that of AgNW film (Appendix A). It is confirmed that the introduction of Mxene improved the adhesion between the substrate and the film. As expected, the introduction of Mxene and interface welding effect enhances the strength of the bond, the adhesive force between the film and the substrate, and reduces the stress on the substrate, thus improving mechanical durability.

The mechanical flexibility of TCFs is also an important performance parameter in practical applications. The bending ratio is defined as (L_0_ − L)/L_0_, where the L_0_ is the pristine length of the film and L is the length after bending. As shown in Figure 5a, TCFs exhibited a stable conductivity under an 80% bending ratio, and the resistance increased by 13% even under a 100% bending ratio. Moreover, the bending cycle test for AgNW and AgNW/MXene TCFs was performed with a bending radius of 8 mm, as shown in Figure 5b. It indicated that the resistance of the AgNW/MXene hybrid film increased by 54% after bending cycles. In contrast, the pristine AgNW film soared dramatically by 205%. These results proved that the AgNW/MXene hybrid film exhibits better mechanical stability.

In addition to the mechanical stability, the environmental stability of AgNW-based TCFs is still a challenge. Thus, we tracked the conductivity of AgNW and AgNW/MXene2 TCFs under various conditions to investigate the stability. The resistance change can be described by RC = R/R_0_, where R is the time-dependent sheet resistance and R_0_ is the initial sheet resistance. All the sheet resistance values are the average of three measured samples. Figure 6a showed the relationship between the resistance of the transparent film and the temperature when annealed at 100 °C. It is observed that the hybrid TCFs exhibited better thermal stability than AgNW TCFs. After 50 h, the sheet resistance of the pristine AgNW TCFs increased to 3.26 times. In contrast, the sheet resistance of AgNW/MXene2 TCF increased by 0.37 times, which illustrated that the hybrid film showed good protection against thermal damage of AgNWs. Within the first hour of annealing, the inset image showed that the sheet resistance of the transparent film first decreases in 15 min, which is considered to be the result of the thermal welding effect. Moreover, the sheet resistance of the pristine AgNW film increased faster than that of the hybrid film when annealed at 180 °C (Appendix A).

To study the storage stability, the AgNW/MXene and pristine AgNW films were exposed to air in an environment at room temperature (~25 °C) and with a humidity of ~43%, and the results are shown in Figure 6b. After 40 days in the ambient environment, the sheet resistance of the pristine AgNW TCFs increased by 466%, whereas that of the AgNW/MXene2 TCFs increased by 38%. The SEM images demonstrated many nanoscale particles are distributed along the silver nanowires and some nanowires have started to fracture, resulting in a rapid increase in the sheet resistance of the AgNWs film (Figure 6c). However, the surface morphology of hybrid film has almost no change compared with that of the previous (Figure 6d). It proved that MXene nanosheets on the surface isolate the AgNW from the corrosive components in the air, thus protecting the AgNW films from degradation. Therefore, the AgNW/MXene2 hybrid TCFs exhibited a more stable conductivity in the ambient environment for a long time.

## 4. Conclusions

AgNW/MXene hybrid TCFs were prepared through an all-solution process; small and large several-layer MXene nanosheets were introduced to AgNW conductive networks by two coatings. MXene sheets tracked along the nanowires can directionally patch the silver nanowires and weld the wire junctions by the capillary effect, which can significantly decrease the contact resistance. In addition, the nanosheets were filled in the voids of the silver nanowire network and covered the AgNW networks. The resulting electrical bridge effect provided additional electronic paths for the hybrid film, thus effectively improved its optoelectronic properties. The optimized AgNW/MXene TCFs displayed a sheet resistance of 15.1 Ω/sq and a transmittance of 89.3% at 550 nm, whose photoelectric performance was comparable to ITO. Moreover, the abundant surface functional groups of MXene improved the hydrophilicity, work function, and adhesion to the substrate of the hybrid films. Moreover, the MXene can block the oxidation of the silver nanowires, thereby effectively improving the long-term stability. The hybrid films maintain remarkable performance after 1600 mechanical bending cycles, annealing at 100 °C for 50 h, and exposure to ambient air for 40 days. As a result, the AgNW/MXene hybrid TCFs prepared by continuous coatings can be expanded in a large area and has great potential to be applied in flexible electronic devices.

## Figures and Tables

**Figure 1 nanomaterials-11-01360-f001:**
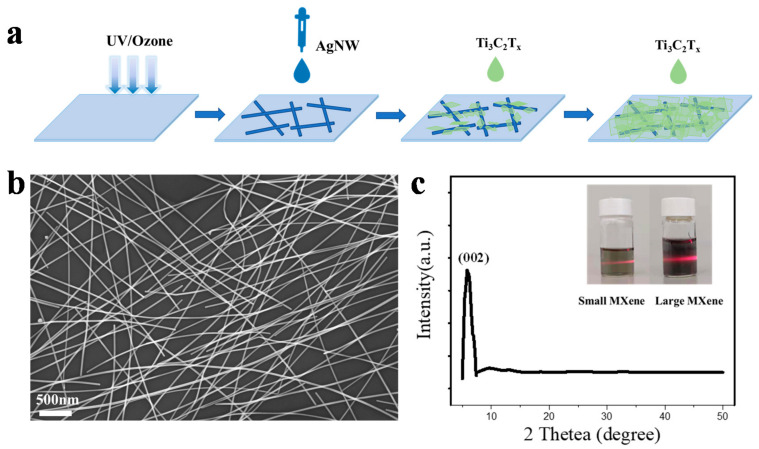
(**a**) Schematic illustration of the AgNW/MXene hybrid transparent conductive films. (**b**) SEM image of AgNWs network. (**c**) XRD patterns for MXene and the graph of Tyndall scattering phenomenon for MXene dispersion (inset).

**Figure 2 nanomaterials-11-01360-f002:**
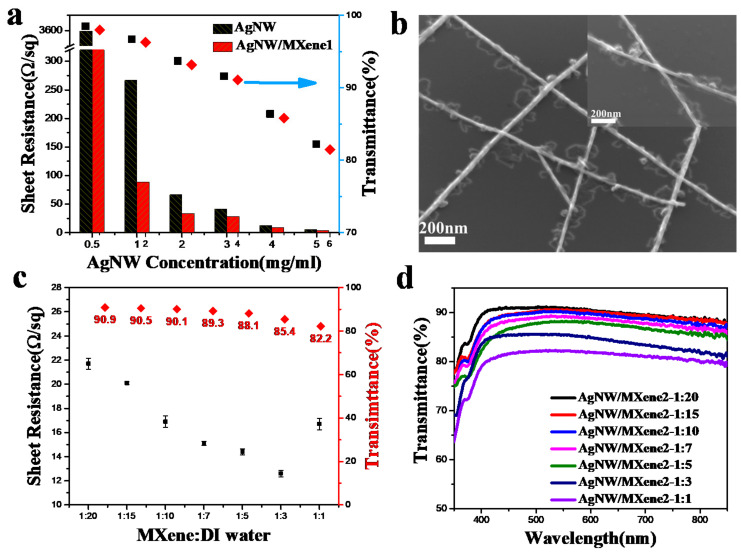
(**a**) The photoelectric properties of pristine AgNWs film and hybrid films with different AgNW concentrations. (**b**) Low-and high-magnification SEM images of the Ti_3_C_2_T_x_ nanosheets on the AgNW network (AgNW/MXene1). (**c**) Sheet resistances and transmittances of the hybrid films under different MXene contents after second coating (AgNW/MXene2). (**d**) Optical transmittance of AgNW/MXene2 hybrid TCFs with various contents.

**Figure 3 nanomaterials-11-01360-f003:**
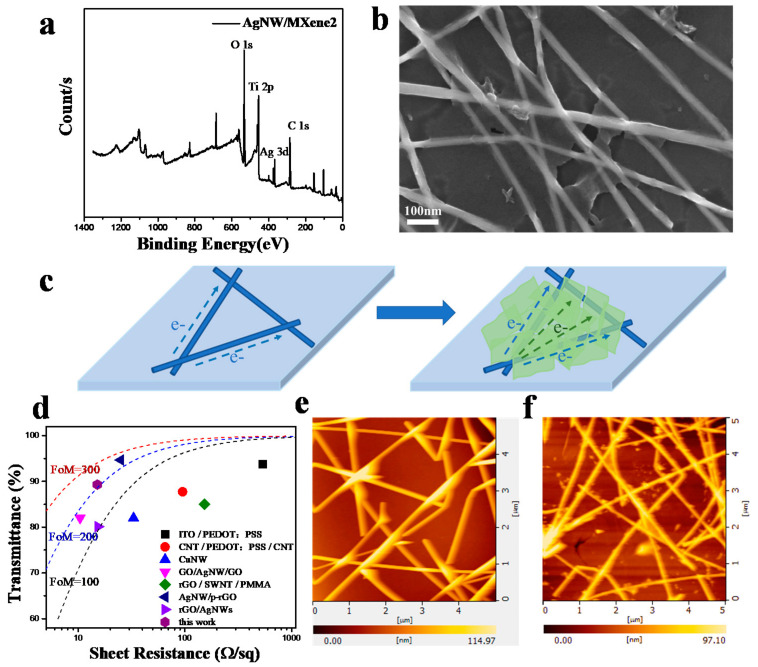
(**a**,**b**) XPS, SEM images of the AgNW/MXene2 hybrid films. (**c**) Schematic diagram of bridge effect. (**d**) Comparison of the optoelectronic performance of the AgNW/MXene electrode with those in previous works. (**e**,**f**) AFM images of the AgNW and AgNW/MXene2 TCFs.

**Figure 4 nanomaterials-11-01360-f004:**
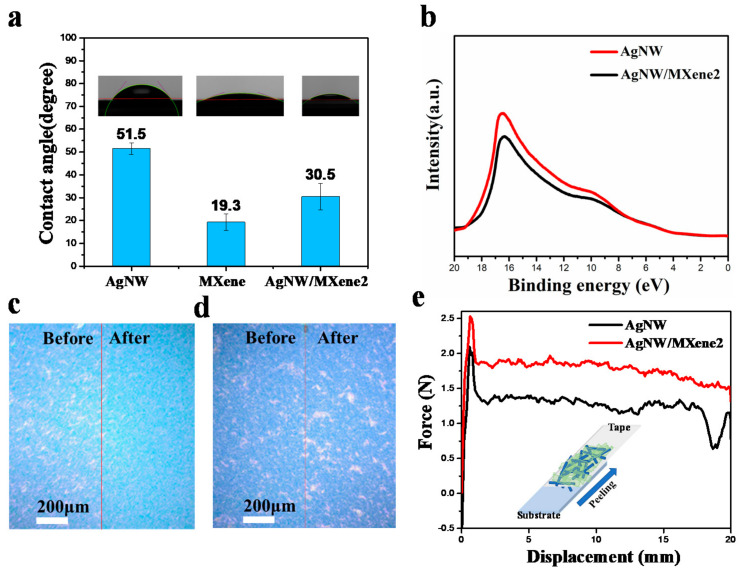
(**a**) The water contact angle values of various TCFs. (**b**) UPS spectra of AgNW and AgNW/MXene2 TCFs. (**c**,**d**) Optical images of AgNW and AgNW/MXene2 film after tapping tests. (**e**) Force–displacement curves for TCFs with and without MXene on PEN substrates.

**Figure 5 nanomaterials-11-01360-f005:**
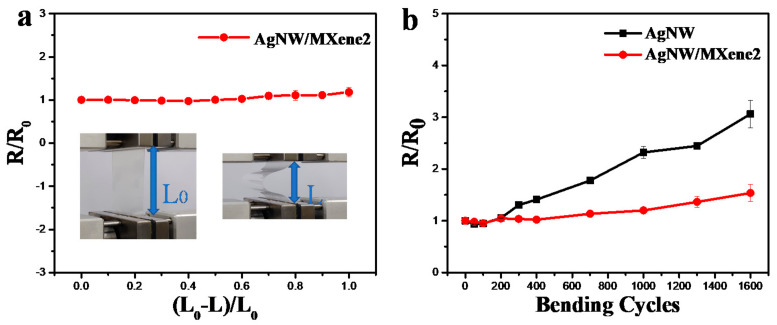
(**a**) Change in resistance of AgNW/MXene TCF to different bending ratios. (**b**) R/R_0_ of AgNW/MXene films as a function of bending cycles.

**Figure 6 nanomaterials-11-01360-f006:**
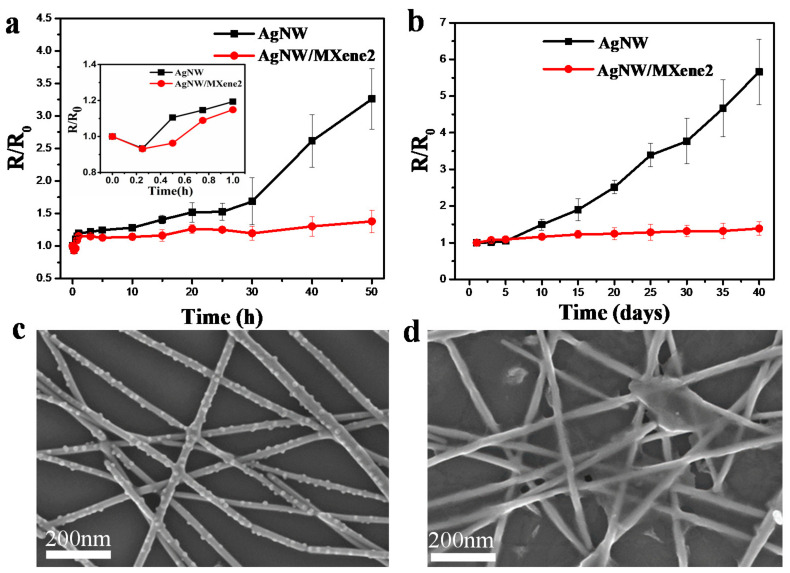
(**a**) Sheet resistance as a function of time at 100 °C on a hotplate, and the curve in the first 1 h (inset). (**b**) Sheet resistance as a function of time in the ambient environment for 40 days. (**c**,**d**) SEM images of pristine AgNW and AgNW/MXene2 TCFs after exposed to air for 40 days.

## Data Availability

The data generated during and/or analyzed during the current study are available from the corresponding authors on reasonable request.

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
