# Peer review of "Scalable Solution-Processed Fabrication Approach for High-Performance Silver Nanowire/MXene Hybrid Transparent Conductive Films"

_nanomaterials, 2021, doi:10.3390/nano11061360_

Round 1

Reviewer 1 Report

This paper is another iteration on Ref. 28. The work does not meet the standards of many of the cited papers (in higher impact journals than Nanomaterials). However, the work might be acceptable for publication in Nanomaterials subject to revision. For example, the XRD is very low quality, as is the AFM. These two features should really be improved upon. Also, the Tyndall effect has nothing to do with the images in Fig. 1 c. There is no variation in the incident wavelength here. All the authors are reporting is light scattering. This is an erroneous statement in Ref. 29.

Author Response

Dear reviewer,

    Thank you very much for taking the time to review our manuscript and submit review comments. We also wish to take this opportunity to thank the reviewer for the constructive comments and valuable recommendations. We have carefully revised the manuscript according to the reviewer’s suggestions.

    At last, please allow us to express our appreciation to your review and suggestions for improvement of our manuscript. We sincerely hope that the revised manuscript is now suitable for publication, if still not, one more chance to improve the quality of manuscript will be greatly appreciated.

Yours Sincerely,

Lianqiao Yang

Reviewer 2 Report

The article shows very interesting results. The research was well planned, and the interpretation of the obtained results also did not raise any objections. I think that the manuscript can be qualified for publication after small corrections. Below are my little comments:

Introduction - I suggest in the table to compare the advantages and disadvantages of the materials used as TCF. This will make it easier for the reader to follow this work.

Introduction - emphasize the scientific novelty of this work

Results and Discussion - equation 1, why is the optical transmission value measured at 550 nm?

Results and Discussion - figure 6b - what was the air humidity? Were tests carried out with variable air humidity?

Conclusion - there is a lack of information and comparisons with the research of other authors. Were similar results obtained?

Author Response

(The authors gave the same response as above.)

Reviewer 3 Report

Dear Authors,

the following points should be added/changed in your manuscript to improve it:

  • Generally, please add the missing spaces between numbers and "large" units (such as nm, h, Ohm, °C etc.). The unit of the sheet resistance is Ohm, not "Ohm/sq". I know that this pseudo-unit is often used, but repeating an error doesn't make it right.
  • Introduction: "Silver nanowire ... high photoelectric performance" - probably you mean something else here.
  • Fig. 1c and corresponding text: Please explain the Tyndall scattering phenomenon in brief - what would be expected for aggregations? Which of the two glasses is the relevant one?
  • Figs. 2a and c: Please add the error bars; the exact values of the sheet resistance are not necessary. What does the arrow in Fig. 2d mean?
  • Fig. 3d: What do the dashed lines mean?
  • Fig. 1: By dividing electric conductivity by optical transmittance, you get a value with the unit 1/Ohm or 1/(Ohm %). So the FoM cannot be a pure number.
  • Fig. 4a: Please add the error bars.
  • Fig. 4e: The arrow shows in a wrong direction for the normal 180° peeling test.
  • Fig. 5, 6a, b: error bars, please.

Author Response

Dear editors and reviewers,

Thank you very much for taking the time to review our manuscript and submit review comments. We also wish to take this opportunity to thank the reviewer for the constructive comments and valuable recommendations. We have carefully revised the manuscript according to the reviewer’s suggestions.

At last, please allow us to express our appreciation to your review and suggestions for improvement of our manuscript. We sincerely hope that the revised manuscript is now suitable for publication, if still not, one more chance to improve the quality of manuscript will be greatly appreciated.

Yours Sincerely,

Lianqiao Yang
